# Supersonic Motion of Atoms in an Octahedral Channel of fcc Copper

**DOI:** 10.3390/ma15207260

**Published:** 2022-10-18

**Authors:** Ayrat M. Bayazitov, Dmitry V. Bachurin, Yuri V. Bebikhov, Elena A. Korznikova, Sergey V. Dmitriev

**Affiliations:** 1Institute of Molecule and Crystal Physics, Ufa Federal Research Center, Russian Academy of Sciences, 450075 Ufa, Russia; 2Research Laboratory for Metals and Alloys under Extreme Impacts, Ufa State Aviation Technical University, 450008 Ufa, Russia; 3Polytechnic Institute (Branch) in Mirny, North-Eastern Federal University, 678170 Mirny, Russia; 4Center for Design of Functional Materials, Bashkir State University, 450076 Ufa, Russia

**Keywords:** fcc lattice, octahedral channel, lattice defects, molecular dynamics simulation, mass transfer, crowdion

## Abstract

In this work, the mass transfer along an octahedral channel in an fcc copper single crystal is studied for the first time using the method of molecular dynamics. It is found that the initial position of the bombarding atom, outside or inside the crystal, does not noticeably affect the dynamics of its motion. The higher the initial velocity of the bombarding atom, the deeper its penetration into the material. It is found out how the place of entry of the bombarding atom into the channel affects its further dynamics. The greatest penetration depth and the smallest dissipation of kinetic energy occurs when the atom moves exactly in the center of the octahedral channel. The deviation of the bombarding atom from the center of the channel leads to the appearance of other velocity components perpendicular to the initial velocity vector and to an increase in its energy dissipation. Nevertheless, the motion of an atom along the channel is observed even when the entry point deviates from the center of the channel by up to 0.5 Å. The dissipated kinetic energy spent on the excitation of the atoms forming the octahedral channel is nearly proportional to the deviation from the center of the channel. At sufficiently high initial velocities of the bombarding atom, supersonic crowdions are formed, moving along the close-packed direction $\langle \bar{1}10\rangle$, which is perpendicular to the direction of the channel. The results obtained are useful for understanding the mechanism of mass transfer during ion implantation and similar experimental techniques.

## 1. Introduction

The mass and energy transfer phenomena are associated with processes such as severe plastic deformation [1], plasma and laser treatment [2,3,4], ion implantation [5], and irradiation [6,7,8,9,10,11]. Ion implantation can be used to dope elements onto a surface [12] and modify the near surface layers to toughen materials [13]. Bioactivity of titanium surface can be promoted by magnesium ion implantation [14]. Atomic-size spin defects can be introduced in diamond by low-energy ion implantation [15]. Amorphization of the diamond lattice by carbon ion implantation takes place after 16% of the atoms are removed from their lattice sites [16]. Ion implantation is an efficient tool for doping graphene [17,18,19]. Ion implantation improves the cutting ability of tools [20,21]. Ion beams are used for polishing samples for optics applications [22]. Implantation of helium ions into Si single crystal was studied by using a molecular dynamics method to analyse the interaction mechanism of helium ions with Si and helium ion migration [23].

High-energy particles, when interacting with a material, cause the appearance of abnormally high concentration of point defects (vacancies and interstitial atoms), which are also called Frenkel pairs, as well as the excitation of nonlinear vibrational atomic modes. The migration activity of interstitial atoms is much higher than that of vacancies, which is related to a lower migration barrier. The formed vacancies and interstitials exist in different configurations, which differ from each other in the degree of spatial localization and the structure of clusters. An interstitial atom in an fcc metal has two stable non-equivalent positions called octahedral and tetrahedral sites. In addition, interstitial can be embedded into a close-packed atomic row. In this case, one speaks of the formation of a crowdion, which has a very low migration energy and, as a consequence, high mobility and therefore plays an important role in mass transfer and structure evolution [24].

Recently, in addition to ‘classical’ 1-crowdion, the so-called *N*-, *M*-, and M×N-crowdions have been intensively studied [25,26,27,28,29,30,31,32]. In the first case, the initial momentum is imparted to *N* neighboring atoms located in the same close-packed row along the row; in the second case, to a group of atoms in *M* neighboring atomic rows; and in the third case, to N×M block of atoms. In particular, it was revealed that the energy necessary for excitation of *N*-crowdion is lower than that of 1-crowdion, but the distance traveled by *N*-crowdion is much longer than that of the ‘classical’ 1-crowdion [33,34,35]. Based on this, the authors [27] concluded that such crowdion clusters can play an important role in mass transfer from the surface into the bulk. Vacancy can also exist in a delocalized form, which is called voidion [29,36,37]. Both interstitial [7,24] and vacancy clusters [38,39] can exhibit cooperative one-dimensional motion. Propagation of M×N-crowdions can be accompanied with a formation of atomic vacancies inside the crystal lattice [29].

Interaction with high-energy particles can also lead to excitation of discrete breathers or localized vibrational modes. The latter are spatially localized high-amplitude vibrational modes in nonlinear lattices, which can be of different dimensionality and can differ in the type of nonlinearity, mobility, excitation methods. The majority of them have been intensively studied over the past decade in various materials with different types of crystal lattice [40,41,42,43,44,45,46,47,48]. Moving discrete breathers, i.e., those that can move along the close-packed atomic row where they were originally excited, can contribute to mass transfer in the material, as well as affect the physical properties of materials [42,49,50,51,52,53,54,55].

When a material surface is bombarded with high energy particles, the latter can experience collisions with surface atoms and can fall into the gaps and channels between the atoms. There are numerous publications devoted to the interaction of particles with a surface, namely when atoms collide head-on, resulting in an excitation of crowdions and a mass transfer along the close-packed crystallographic directions. However, we are not aware of publications considering the motion of a bombarding atom along atomic channels. The aim of the present work is to study the dynamics of bombarding an atom in an octahedral channel of fcc copper by means of molecular dynamics simulation. The impact of impurity atoms, point or extended defects, stress fields, etc., which obviously can affect the dynamics of the process, is not considered in this work.

## 2. Materials and Methods

Molecular dynamics simulations are based on a solution of Newtonian equations of motion and are performed via the Large-Scale Atomic/Molecular Massively Parallel Simulator (LAMMPS) [56]. Empirical many-body interatomic potential for fcc copper based on the embedded atom method was adopted for modeling [57]. This potential was well tested for modeling the structure and properties of copper and its alloys, and due to its relative simplicity, it was not demanding on computing power. The equilibrium lattice constant at zero temperature reproduced by the potential was *a* = 3.615 Å, and the interatomic distance was *d* = *a*/$\sqrt{2}$ = 2.556 Å.

The elementary translational cell contained 8 atoms and was chosen so that the *x*, *y*, and *z* axes coincided with the crystallographic directions 〈110〉, $\langle \bar{1}10\rangle$, and 〈001〉, respectively. The octahedral pore channels were perpendicular to the crystal surface and oriented along the *x* axis. The number of unit cells along the *x*, *y*, and *z* axes were set to be 25, 60, and 50, which was enough for this type of simulation. The total number of atoms in the computational cell was $6\times 10^{5}$, and its volume was 127.90×306.8×180.4 Å${}^{3}$. Periodic boundary conditions were employed along the *y* and *z* coordinate directions. The computational cell had a vacuum layer with the thickness of 102.26 Å along the *x* axis. Thus, a monocrystalline sample containing two free surfaces parallel to the plane (y,z) was modeled.

The bombardment of the crystal surface by copper atoms was simulated. In order to find out how the initial location of the bombarding atom affects the dynamics of its further movement in the crystal, two scenarios were considered. In the first scenario, the atom was located in the vacuum at a distance of *h* = 6.94 Å from the free surface and moved with an initial velocity $V_{0x}$ towards the crystal. In the second scenario, the atom started its motion inside the crystal (under the free surface) at two different depths of $h_{1}$ = 8.542 Å and $h_{2}$ = 23.841 Å with a given initial velocity. Preliminary calculations demonstrated that the *x*-component of the initial velocity of the bombarding atom should be in the range of 200 to 500 Å/ps, since at lower velocity values, the bombarding atom cannot overcome the potential barrier and penetrate deep into the crystal through the pore channel. The stability of the motion of the bombarding atom in octahedral pore channel was checked by introducing small velocity components $V_{y}=V_{z}=10^{-5}$ Å/ps along the *y* and *z* axes (normal to the channel axis).

To establish how the position of the entry point of the bombarding atom, with respect to the octahedral channel, affects its ability to penetrate the crystal, the following four cases were considered. In case I, the point of entry of the bombarding atom (shown in red in Figure 1a) was located exactly in the center between the four neighboring surface atoms, in other words, in the center of the channel. In case II, the bombarding atom was shifted along the positive direction of the *y* axis by some distance, with respect to the center. In case III, the bombarding atom had equal displacement components along the both *y* and *z* axes. In case IV, the bombarding atom was shifted only along the *z* axis (see Figure 1a).

The molecular dynamics simulations were performed at the temperature of 300 K via the NVE thermodynamic ensemble (the constant number of atoms, the volume of the computational cell, and the total energy of the system). The equations of motion of the atoms were integrated using the Verlet method of the fourth order of accuracy, with a time step of 1 fs.

Potentials with Ziegler-Biersack-Littmarck (ZBL) hardening [58] are often used to model collision cascades in materials irradiated with swift ions. The ZBL correction of the repulsive part of the potential becomes important at ion energies above 10 keV [59,60,61,62]. In our simulations, the highest copper ion velocity considered was 500 Å/ps, which corresponds to a kinetic energy of 800 eV or 12.5 eV per nucleus. Therefore, there is no need to use the ZBL correction for such low ion energies.

The following measures were taken to control simulation errors. The size of the computational cell was chosen to be large enough so that the perturbations caused by the bombarding atom did not reach the cell boundaries during the simulation. As mentioned above, a time step of 1 fs was used to integrate the equations of atomic motion. Reducing the time step to 0.5 fs had no noticeable effect on the simulation results. Thermal fluctuations introduced stochasticity into the motion of atoms. Three runs were performed for each set of simulation parameters to see the effect of thermal fluctuations. For example, for $V_{0}=250$ Å/ps, in three realizations, the penetration depth of the bombarding atom was 17, 18, and 22 Å with a mean value of 19 Å and a standard deviation of 2.1 Å. For $V_{0}=450$ Å/ps, the penetration depth in three realizations was 104, 107, and 112 Å, with an average value of 108 Å and a standard deviation of 3.3 Å. It can be concluded that the considered temperature of 300 K does not give a large scatter of numerical data.

## 3. Results and Discussion

### 3.1. Influence of Initial Position of a Bombarding Atom with Respect to the Crystal

As mentioned above, two scenarios of the initial location of the bombarding atom were considered. The relative change in the *x*-component of the velocity of the bombarding atom as a function of time for the both scenarios is presented in Figure 2a. In the first scenario, i.e., when the bombarding atom is outside the crystal at a distance of *h* = 6.94 Å from the free surface, the change in the relative velocity began only at the time t≈0.02 ps, which was due to the fact that this time was required for the bombarding atom to approach the free surface of the crystal. After that, the relative velocity began to reduce and change in a wave-like manner. The latter was associated with overcoming potential barriers when the atom moved along an octahedral channel in the crystal. In the second scenario, the change in the relative velocity started immediately after the onset of the simulation. In both scenarios studied, the bombarding atom overcame 12 potential barriers at the initial velocity of 300 Å/ps. As clearly seen in Figure 2a, the depth at which the bombarding atom was located at the initial moment of time did not have any effect on the dynamics of its further movement inside the channel. If we remove the horizontal section of the curve, which corresponds to the motion of an atom outside the crystal for the first scenario, and then shift the blue curve to the left, then it almost exactly coincides with the corresponding curves for the second scenario. The latter fact suggests that the initial position of the atom, from which the initial velocity is set when entering the octahedral channel, did not have any significant effect on its further motion. Therefore, further in the work, only the first scenario was considered, i.e., the bombarding atom was initially located in vacuum at some distance from the free surface.

### 3.2. Influence of the Location of the Entry Point of the Bombarding Atom

The four different cases of location of the entry point of the bombarding atom, with respect to the octahedral channel, were considered, as described in Section 2.

The results for case I, when the atom hit the surface exactly in the center of the channel and had initial velocities in the range from 200 to 500 Å/ps, are shown in Figure 2b. The *x*-component of the atom’s velocity was normalized to the initial velocity $V_{0x}$. As long as the atoms moved in vacuum, its velocity was constant and started to decrease as the atom entered the octahedral channel. The curves oscillated due to the fact that the atom moved in the periodic potential created by the surrounding atoms, and the step of the potential was equal to the interatomic distance *d*. The number of maxima on the curves indicates the depth of penetration of the atom deep into the crystal. Note that an atom launched with $V_{0x}=200$ Å/ps (green curve in Figure 2b) traveled only three interatomic distances and was trapped by the lattice. As $V_{0x}$ increased, the penetration depth increased, and at $V_{0x}=500$ Å/ps, the atoms traveled in the channel many tens of interatomic distances.

The dependence of different velocity components for the bombarding atom launched with the initial velocities 300 and 450 Å/ps for case II is demonstrated in Figure 3. The displacement along the *y* axis, $\Delta_{y}$, is in the range of 0.1–0.5 Å. In this case, the penetration depth of the bombarding atom increased with an increase in initial velocity. Increasing the displacement relative to the center of the channel significantly reduced the velocity of the atom. Moreover, the higher the initial displacement of the atom along the *y* axis, the higher the acquired transverse velocity component $V_{y}$, and the greater the decrease in its initial velocity, i.e., $V_{x}$ component. The bombarding atom also acquired a $V_{z}$ velocity component. However, at the initial stage, it was noticeably lower than $V_{y}$. In addition, the transverse velocity components, $V_{y}$ and $V_{z}$, decreased significantly over time, which indicates that the atom stabilizes at the center of octahedral channel and prefers to be equidistant from the atomic lens consisting of four atoms. However, the moving atom spent its energy on excitation of transverse velocity components and, as a result, slowed down faster compared to case I. An increase in the initial velocity of the bombarding atom lead to an increase in the penetration depth into the material.

In case III, i.e., when the bombarding atom was simultaneously displaced relative to the *y* and *z* axes, the deceleration of the atom occurred faster than in case II, as depicted in Figure 4. The resulting transverse velocity components, $V_{y}$ and $V_{z}$, were closer in absolute value. The greater the displacement relative to the center of the octahedral channel, the faster the atom lost its velocity $V_{x}$ and finally stopped. The bombarding atom spent its kinetic energy on the excitation of two components $V_{y}$ and $V_{z}$. The atom lost its initial velocity much faster than in case II. In addition, since the distance between the close-packed atomic rows forming the octahedral channel was greater along the *z* axis than that along the *y* axis (see scheme in Figure 1a), the initial displacement of the bombarding atom along the *y* axis influenced the further dynamics of the atom more strongly in comparison with the displacement along the *z* axis.

The velocity components of the bombarding atom as functions of time for case IV are displayed in Figure 5. Even relatively large deviations from the center along the *z* axis did not have a significant effect on the dynamics of the atom in the octahedral channel, while significant changes in the dynamics appear only at the initial velocity of 450 Å/ps for the displacement of $\Delta_{y}=$0.5 Å and after 0.15 ps of the simulation time. Similar to the previous cases, the smaller initial deviations of the moving atom from the center of the channel lead to the smaller values of the transverse velocity components, $V_{y}$ and $V_{z}$.

Thus, the consideration of the four cases of the location of the bombarding atom when entering inside the octahedral channel allows us to conclude that the dynamics of its motion with simultaneous displacement along the *y* and *z* axes (case III) has an intermediate character between the cases II and IV. Independently on the location of the bombarding atom, the latter always remains in the channel and cannot leave it. Over time, the velocity of the moving atom decreased to zero, as a result of its interaction with neighboring atoms.

The dependence of the penetration depth of the bombarding atom located at the center of the octahedral channel as a function of its initial velocity is demonstrated in Figure 6. Each point gives a penetration depth value averaged over three numerical runs. The penetration depth increases with an increase in the initial velocity. This dependence is well fitted by a cubic parabola, and therefore it can be concluded that the penetration depth *h* is proportional to the cube of the initial velocity, i.e., $h∼V_{0}^{3}$.

Figure 7 shows the dependence of the velocity of the bombarding atom on its displacement with respect to the channel center. For case II, an increase in the displacement from the center lead to a gradual decrease in the atom’s velocity. Small displacements up to 0.3 Å did not have a noticeable effect on the atom velocity in the channel for case IV, and only at values exceeding 0.3 Å the velocity began to decrease significantly. For case III, even for relatively large initial displacements along the *y* and *z* axes, the velocity of the bombarding atom at the time instant of 0.25 ps did not differ much from the initial one. The exception was the displacement of 0.1 Å, where the velocity decreased more remarkably than at large initial displacements, which was due to the more pronounced influence of neighboring atoms.

When the bombarding atom was close to the atoms of the octahedral channel, that is, at relatively large initial deviations from the channel center, noticeable displacements of the channel atoms from their equilibrium lattice sites took place. The latter lead to an excitation of supersonic crowdions along the *y* axis, i.e., close-packed $\langle \bar{1}10\rangle$ crystallographic direction, as illustrated in Figure 8. When the displacement of the excited atom was more than h/2, it could not move back into the original equilibrium lattice site and shifted into the neighboring one, which corresponded to crowdion motion. Moreover, at high initial velocities of the bombarding atom, multiple crowdions can be formed. The formation of crowdions precisely in the indicated crystallographic direction was due to the fact that for a given orientation of the single crystal, these close-packed directions were the most favorable and turned out to be perpendicular to the initial velocity vector of the bombarding atom. The average velocities of the crowdions marked as $C_{1}$, $C_{2}$, $C_{3}$ in Figure 8 were 121.2, 95.4, and 105.0 Å/ps, respectively. The speed of sound along the $\langle \bar{1}10\rangle$ crystallographic direction in fcc copper modelled with the used interatomic potential was 38 Å/ps, which allows us to state that supersonic crowdions are excited. At lower initial velocities of the bombarding atom, its kinetic energy was insufficient to overcome the potential barrier. In the latter case, no formation of the Frenkel pair occurred and only propagation of compressive soliton (also called focuson) was observed. Supersonic 1-crowdions and focusons were studied earlier in a number of works [33,34].

## 4. Conclusions

The dynamics of the bombarding atom inside the octahedral channel in a single crystal of fcc copper was investigated using molecular dynamics simulation. The following main conclusions can be drawn. The initial position of the bombarding atom (outside or inside the crystal) did not have a significant effect on the dynamics of its motion. The depth of penetration of the bombarding atom into the material was proportional to the cube of initial velocity. The smallest dissipation of kinetic energy, or the greatest penetration depth, had the atom moving exactly in the center of the octahedral channel. The deviation of the bombarding atom from the center of the channel lead to the appearance of transverse velocity components. The greater the deviation of the atom from the center of the channel, the greater the dissipation of kinetic energy on the neighboring atoms. On the other hand, even with a relatively large deviation from the center of the octahedral channel (up to 0.5 Å), a high-energy bombarding atom can penetrate many tens of interatomic distances into the lattice. At high initial velocities of the bombarding atom, supersonic crowdions were excited along the close-packed $\langle \bar{1}10\rangle$ direction perpendicular to the initial velocity vector. The excitation of such crowdions is the mechanism of energy dissipation into the lattice from the bombarding atom.

The main difference between mass transfer by atoms moving in a pore channel and transfer by crowdions is that the bombarding atom in the first case penetrates deep into the lattice, and in the second case, it sticks to the surface and produces a relay-race motion of interstitial atoms of the target.

In this work, to avoid the problem of selecting interatomic potentials, the bombarding atom was chosen to be the same as the target atoms (copper), but in future studies, it is important to analyze the cases when the bombarding atom is smaller or larger than the target atoms.

The results of this study are important for the analysis of atomic collisions during surface modification by ion implantation.

## Figures and Tables

**Figure 1 materials-15-07260-f001:**
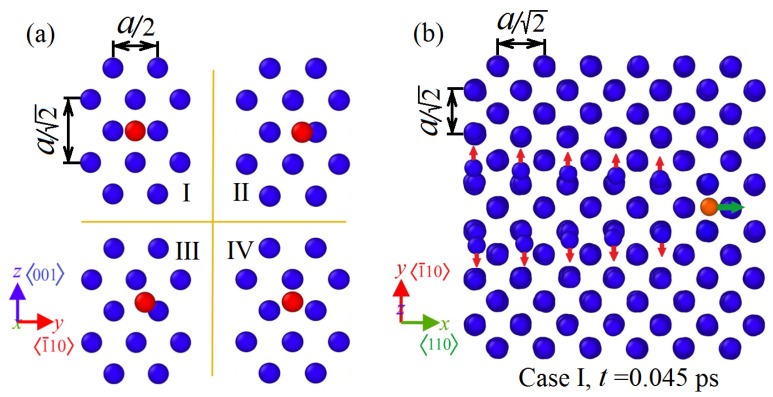
(**a**) Schematic illustration of the four considered entry points of the bombarding atom (highlighted in red) into the crystal. The *x*, *y*, and *z* axes are directed along the crystallographic directions 〈110〉, $\langle \bar{1}10\rangle$, and 〈001〉, respectively. (**b**) The snapshot demonstrating motion of the bombarding atom in the octahedral pore channel along the *x*-direction. The velocity vector of the moving atom is shown by the green arrow. The atoms behind the moving bombarding atom obtain momentum in the direction perpendicular to the velocity vector of the bombarding atom (shown by the red arrows). A snapshot of the structure at time t=0.045 ps for case I is presented.

**Figure 2 materials-15-07260-f002:**
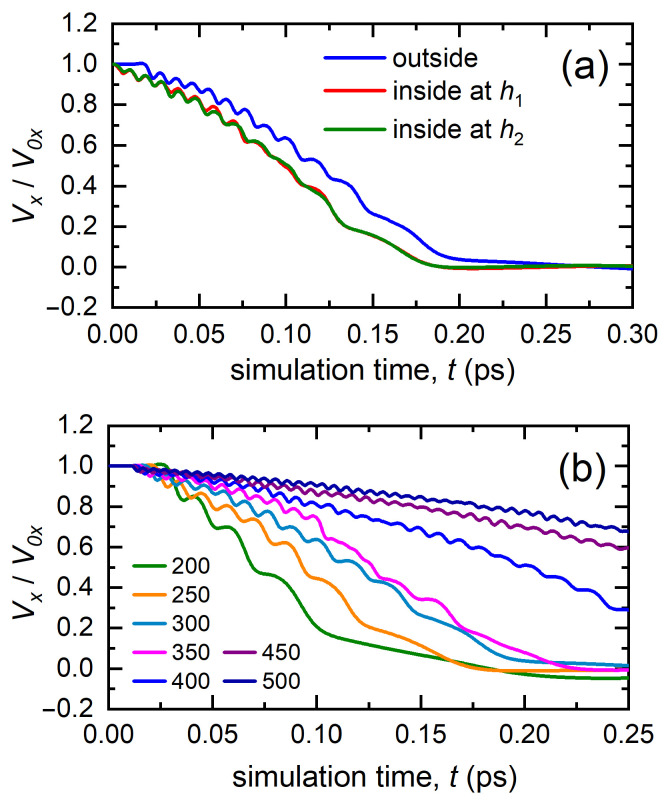
The *x*-component of the velocity of the bombarding atom as a function of simulation time. The velocity is normalized to its initial value $V_{0x}$. (**a**) The case of different initial positions of the bombarding atom launched with the initial velocity of 300 Å/ps. The blue line corresponds to the case when the atom is located outside the crystal at the distance of *h* = 6.94 Å from the free surface (marked as ‘outside’ in the legend for Figure 2a). The red and green lines correspond to the initial position of the atom inside the crystal at the depth of $h_{1}$ = 8.542 Å (marked as ‘inside at $h_{1}$′ in the legend) and $h_{2}$ = 23.841 Å (marked as ‘inside at $h_{2}$′). In all cases, the atom starts at the axis of the octahedral channel. (**b**) The case of different initial velocities of the bombarding atom, excited according to scenario two, case I.

**Figure 3 materials-15-07260-f003:**
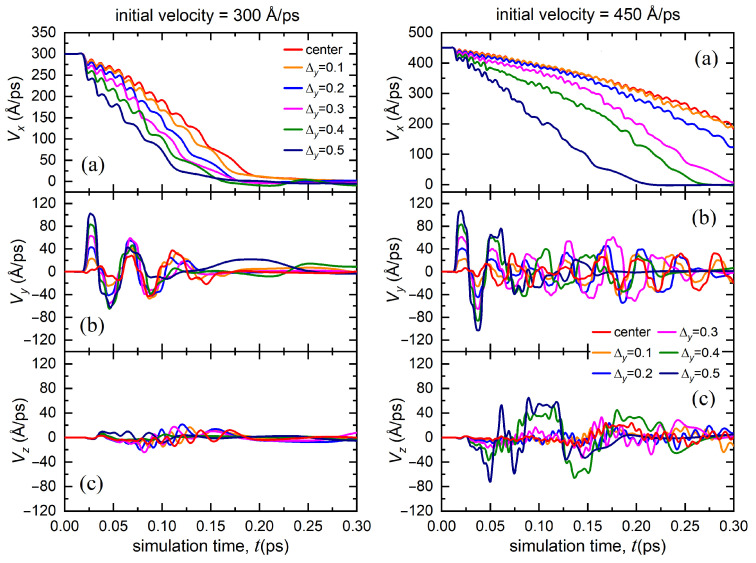
Dependence of the velocity components of the bombarding atom (**a**) $V_{x}$, (**b**) $V_{y}$, (**c**) $V_{z}$ on the simulation time calculated for various displacements of the atom along the *y* axis from the center of the octahedral channel (case II). The results are presented for two values of the initial velocity of the bombarding atom excited according to scenario two: 300 Å/ps (left panel) and 450 Å/ps (right panel).

**Figure 4 materials-15-07260-f004:**
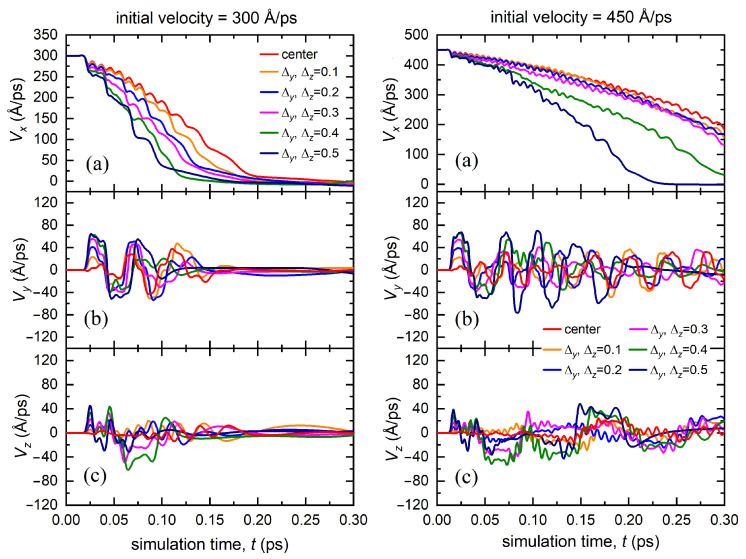
Dependence of the velocity components of the bombarding atom (**a**) $V_{x}$, (**b**) $V_{y}$, (**c**) $V_{z}$ on the simulation time calculated for various displacements of the atom along the *y* and *z* axes from the center of the octahedral channel (case III). The results are presented for two values of the initial velocity of the bombarding atom excited according to scenario two: 300 Å/ps (left panel) and 450 Å/ps (right panel).

**Figure 5 materials-15-07260-f005:**
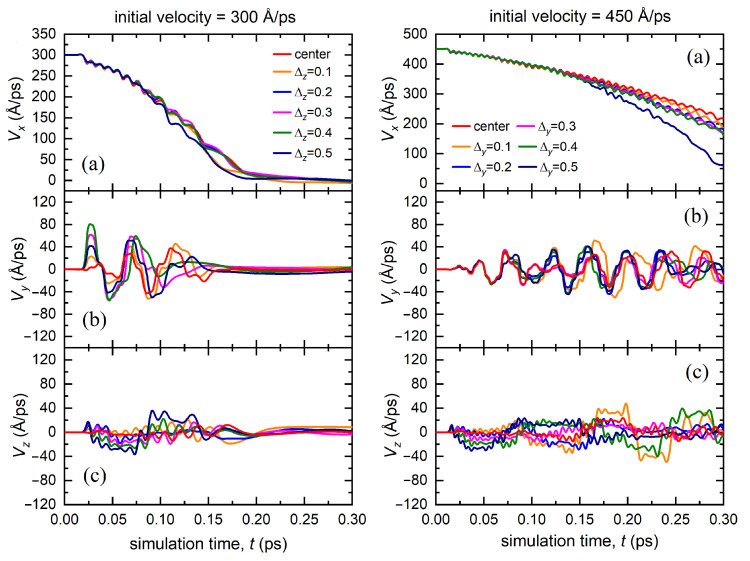
Dependence of the velocity components of the bombarding atom (**a**) $V_{x}$, (**b**) $V_{y}$, (**c**) $V_{z}$ on the simulation time calculated for various displacements of the atom along the *z* axis from the center of the octahedral channel (case IV). The results are presented for two values of the initial velocity of the bombarding atom excited according to scenario two: 300 Å/ps (left panel) and 450 Å/ps (right panel).

**Figure 6 materials-15-07260-f006:**
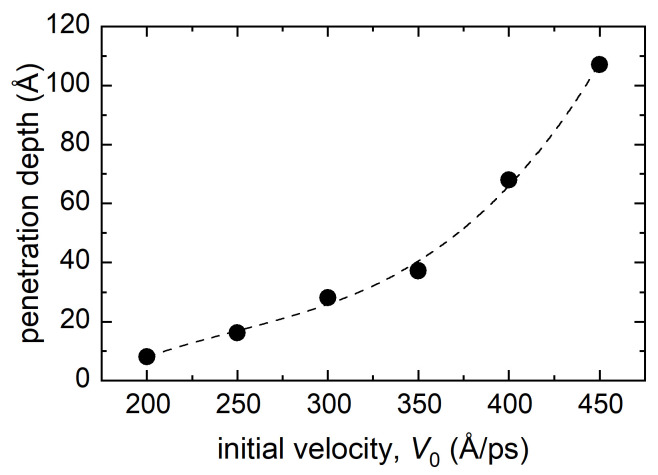
Dependence of penetration depth of the bombarding atom located at the center of the octahedral channel (case I) at different initial velocities. The dashed line depicts the fit with a cubic polynomial.

**Figure 7 materials-15-07260-f007:**
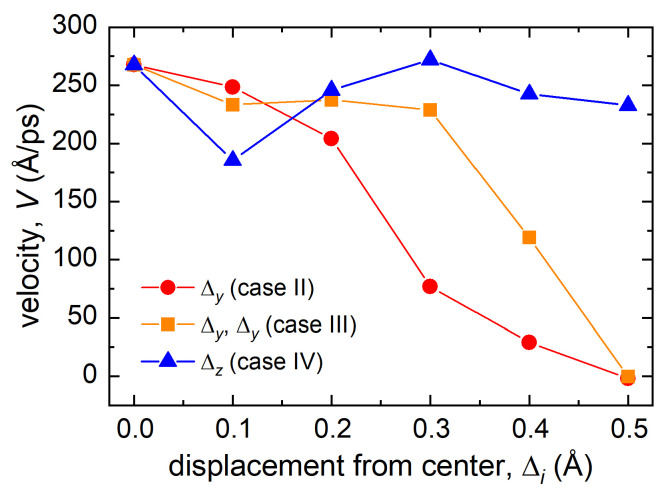
Dependence of velocity of the bombarding atom on the displacements of the atom (cases II–IV). The results are presented for the time instant of *t* = 0.25 ps and the initial velocity of 450 Å/ps. The lines connecting the data points are guides to the eye.

**Figure 8 materials-15-07260-f008:**
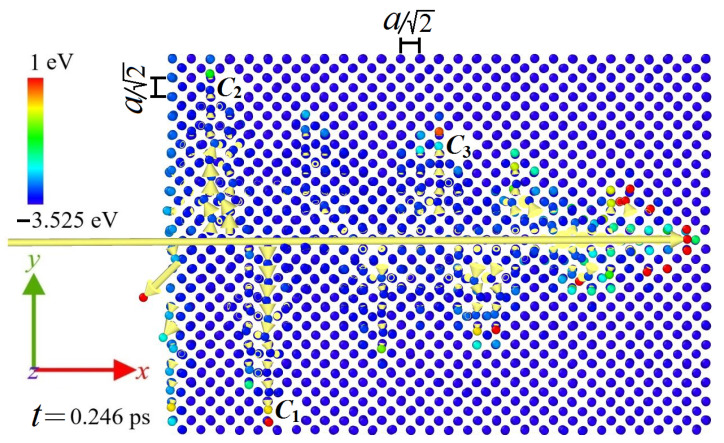
The dynamics of the bombarding atom in fcc copper along the octahedral channel, i.e., along the *x* axis (shown by a horizontal yellow arrow). The short yellow arrows along the *y* axis demonstrate the direction of movement of the excited crowdions $C_{1}$, $C_{2}$, and $C_{3}$. The results are presented for the time instant of *t* = 0.246 ps. For this particular case, $V_{x}$ = 450 Å/ps and $\Delta_{y}$ = 0.3 Å (case II).

## Data Availability

The raw/processed data required to reproduce these findings can be obtained from the authors upon request.

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
