# Peer review of "Supersonic Motion of Atoms in an Octahedral Channel of fcc Copper"

_materials, 2022, doi:10.3390/ma15207260_

Round 1
Reviewer 1 Report
This is an excellent paper. there are only a few comments as given below.
L63, take out it seems quite obvious that. Figure 1, give scales. Is there any way to estimate error of these calculations? Figure 6 needs a scale.
L171, take out it is for this reason that.
L204, take out Note that. Please quantify the conclusion section.
Author Response
Dear Reviewers,
we are very grateful for detailed reviews of the manuscript "Supersonic motion of atoms in octahedral channel of fcc copper" (materials-1947790). We considered all the comments and made appropriate changes to the manuscript. All important changes are highlighted in the manuscript in red. Please see our response and a description of the changes below.
Response to the First Reviewer
Reviewer 1: This is an excellent paper. there are only a few comments as given below.
L63, take out it seems quite obvious that.
Our response:
We have removed “it seems quite obvious that”.
Reviewer 1: Figure 1, give scales. Is there any way to estimate error of these calculations? Figure 6 needs a scale.
Our response:
We have indicated the distances between the rows of atoms in Fig. 1 and Fig. 8 (former Fig. 6).
Regarding the estimation of the error of our simulations, the following text was added at the end of Sec. 2. Materials and Methods:
The following measures were taken to control simulation errors. The size of the computational cell was chosen large enough so that the perturbations caused by the bombarding atom did not reach the cell boundaries during the simulation. As mentioned above, a time step of 1 fs was used to integrate the equations of atomic motion. Reducing the time step to 0.5 fs had no noticeable effect on the simulation results. Thermal fluctuations introduce stochasticity into the motion of atoms. Three runs were performed for each set of simulation parameters to see the effect of thermal fluctuations. For example, for V0=250 Å/ps, in three realizations the penetration depth of the bombarding atom was 17, 18 and 22 Å with a mean value of 19 Å and a standard deviation of 2.1 Å. For V0=450 Å/ps, the penetration depth in three realizations was 104, 107 and 112 Å with an average value of 108 Å and a standard deviation of 3.3 Å. It can be concluded that the considered temperature of 300 K does not give a large scatter of numerical data.
Reviewer 1: L171, take out it is for this reason that.
Our response:
We have removed “it is for this reason that”.
Reviewer 1: L204, take out Note that.
Our response:
We have removed “Note that”.
Reviewer 1: Please quantify the conclusion section.
Our response:
The following text was added to the Conclusions:
The depth of penetration of the bombarding atom into the material is proportional to the cube of initial velocity.
On the other hand, even with a relatively large deviation from the center of the octahedral channel (up to 0.5 Å), a high-energy bombarding atom can penetrate many tens of interatomic distances into the lattice.
Reviewer 2 Report
The MD method has been used to describe the mass transfer along an octahedral channel in an FCC Cu, and some results have been obtained, however, there are two questions about the article as following:
(1) the authors should give the reliability of the choice of the potential functions and the timestep used in the simulations.
(2) when the entry point deviates from the octahedral center, what is the minimum distance between atoms,the ZBL method should be taken into account?
(3) The authors mention that the dissipated kinetic energy spent on the excitation of the atoms forming the octahedral channel is nearly proportional to the deviation from the center of the channel, and the depth of penetration of the bombarding atom into the material is nearly proportional to its initial velocity. It might be clearer to show the quantitative linear relationship between these parameters through figures.
Author Response
Dear Reviewers,
we are very grateful for detailed reviews of the manuscript "Supersonic motion of atoms in octahedral channel of fcc copper" (materials-1947790). We considered all the comments and made appropriate changes to the manuscript. All important changes are highlighted in the manuscript in red. Please see our response and a description of the changes below.
Response to the Second Reviewer
Reviewer 2: (1) the authors should give the reliability of the choice of the potential functions and the timestep used in the simulations.
Our response:
The following text was added to Sec. 2. Materials and Methods:
This potential is well tested for modeling the structure and properties of copper and its alloys, and due to its relative simplicity, it is not demanding on computing power.
Reviewer 2: (2) when the entry point deviates from the octahedral center, what is the minimum distance between atoms,the ZBL method should be taken into account?
Our response:
The following text was added to Sec. 2. Materials and Methods:
Potentials with Ziegler-Biersack-Littmarck (ZBL) hardening are often used to model collision cascades in materials irradiated with swift ions. The ZBL correction of the repulsive part of the potential becomes important at ion energies above 10 keV. In our simulations, the highest copper ion velocity considered was 500 Å/ps, which corresponds to a kinetic energy of 800 eV or 12.5 eV per nucleus. There is no need to use the ZBL correction for such low ion energies.
Reviewer 2: (3) The authors mention that the dissipated kinetic energy spent on the excitation of the atoms forming the octahedral channel is nearly proportional to the deviation from the center of the channel, and the depth of penetration of the bombarding atom into the material is nearly proportional to its initial velocity. It might be clearer to show the quantitative linear relationship between these parameters through figures.
Our response:
We agree. Two new figures (Figs. 6 and 7) were added. In Fig. 6, the dependence of penetration depth of the bombarding atom located at the center of the octahedral channel (case I) at different initial velocities is presented. We note that the dependence is actually not linear but rather cubic. In Fig. 7, we present the dependence of velocity of the bombarding atom on the displacements of the atom (cases II-IV). The results are presented for the time instant of t=0.25 ps and the initial velocity of 450 Å/ps. The text describing these figures was added.
Round 2
Reviewer 2 Report
I have no any comments in further.